# Antimicrobial Challenge in Acute Care Surgery

**DOI:** 10.3390/antibiotics11101315

**Published:** 2022-09-27

**Authors:** Carlo Alberto Schena, Gian Luigi de’Angelis, Maria Clotilde Carra, Giorgio Bianchi, Nicola de’Angelis

**Affiliations:** 1Unit of Digestive and HPB Surgery, CARE Department, Henri Mondor Hospital, AP-HP, 94010 Créteil, France; 2Gastroenterology and Endoscopy Unit, Department of Medicine and Surgery, University Hospital of Parma, 43126 Parma, Italy; 3Rothschild Hospital, AP-HP, Université Paris Cité, U.F.R. of Odontology, 75006 Paris, France

**Keywords:** infections, acute care surgery, antimicrobial resistance, source control, infection prevention and control, antimicrobial stewardship, systemic inflammatory response syndrome

## Abstract

The burden of infections in acute care surgery (ACS) is huge. Surgical emergencies alone account for three million admissions per year in the United States (US) with estimated financial costs of USD 28 billion per year. Acute care facilities and ACS patients represent boost sanctuaries for the emergence, development and transmission of infections and multi-resistant organisms. According to the World Health Organization, healthcare-associated infections affected around 4 million cases in Europe and 1.7 million in the US alone in 2011 with 39,000 and 99,000 directly attributable deaths, respectively. In this scenario, antimicrobial resistance arose as a public-health emergency that worsens patients’ morbidity and mortality and increases healthcare costs. The optimal patient care requires the application of comprehensive evidence-based policies and strategies aiming at minimizing the impact of healthcare associated infections and antimicrobial resistance, while optimizing the treatment of intra-abdominal infections. The present review provides a snapshot of two hot topics, such as antimicrobial resistance and systemic inflammatory response, and three milestones of infection management, such as source control, infection prevention, and control and antimicrobial stewardship.

## 1. Introduction

Acute care surgery (ACS) is traditionally represented by a triad composed by trauma, emergency general surgery, and surgical critical care [1]. It was conceived with the goal of combining skills from trauma surgeons, emergency surgeons, and intensivists, into a single multifaceted and comprehensive discipline. Later, both elective general surgery and surgical rescue were proposed as two additional pillars of ACS [2].

The epidemiological impact of ACS is important because of the cumulative amount of emergency and trauma surgery related morbi-mortality. The burden of surgical emergencies is high; of the three million emergency admissions per year in the United States (US) approximately 30% requires surgery [3], with 896,000 deaths reported in 2010 [4]. Moreover, trauma constitutes the first cause of death in patients under 44 years of age and the fourth cause in the elderly [5]. The estimated financial burden of emergency surgery in US is USD 28 billion every year and is expected to increase up to USD 41 billion by 2060 [6,7]. Despite clinical and financial improvements after ACS model implementation and diffusion [8], ACS patients continue to represent a high-risk population experiencing poorer outcomes. Indeed, up to one-third of ACS patients present with multiple comorbidities and frailty. Emergency surgery is associated with higher rates of mortality, postoperative complications, hospital readmissions, and costs, when compared to corresponding elective procedures [9,10,11,12]. The incidence of healthcare-associated infections (HAIs) is doubled in patients undergoing surgical procedures [13]. In this complex scenario, intra-abdominal and healthcare-associated infections have a great impact, taking into account the critical conditions leading to ACS and the scarce physiologic reserve of these patients. Thus, a close cooperation between the acute care surgeon and the healthcare system is necessary to put in place a comprehensive evidence-based management of infections in the ACS setting, prevent adverse outcomes, and optimize treatment efficacy.

The aim of the present review is to provide a comprehensive overview on the binomen infections-ACS, passing through the analysis of two brain-tease, as antimicrobial resistance (AMR) and systemic inflammatory response syndrome (SIRS), and three cornerstones of infection management, as source control (SC), infection prevention and control (IPC), and antimicrobial stewardship (AS). 

## 2. Antimicrobial Resistance

AMR is a threatening and far-reaching public-health conundrum amplified by the excessive and improper use of antibiotics in humans, animals, and agriculture [14,15,16,17,18]. It reflects the acquired ability of some bacteria to survive and to adapt in the presence of cytotoxic concentrations of antibiotics, thereby resulting in limited and less effective therapeutic options. Nowadays, the term AMR refers not only to bacteria but also to virus, fungi, and other germs. Antibiotic tolerance can be an attenuated form of AMR, distinguished by reduced therapy effectiveness resulting in a longer time before death [19].

AMR is a natural and inevitable evolution mechanism based on drug resistance promoting genes that have spread in bacterial genomes over the millennia before the dawn of antibiotics about 70 years ago [20,21]. Indeed, the metagenomic analysis of 30.000-year-old permafrost sediments by D’Costa VM et al. documented miscellaneous genes promoting resistance to various antibiotics, such as β-lactams, tetracyclines, and glycopeptide antibiotics, also confirming the analogies between the ancient and modern quaternary and tertiary structure of VanA enzyme [22]. Despite its ancient origins, AMR has increasingly emerged as one of the biggest health threats of the 21st century due to two main factors: first, the overuse and the misuse of antimicrobials has evidently sped up the natural pace of AMR, even through the selection of pre-existing AMR-determinants in the microbial pan-genome; second, the development of new classes of antimicrobial agents from the 1960s until today has suffered from a negative decline, so that modern medical armamentarium results depleted [14,15,20,23,24,25]. This promoted the spread of many antibiotic resistant bacteria, the so-called superbugs, such as Methicillin-resistant *Staphylococcus aureus* (MRSA), vancomycin-resistant enterococci (VRE), drug-resistant Escherichia coli, and many more.

Based on the hypothetical scenario of increasing AMR for six pathogens in the next 30 years, the Review on Antimicrobial Resistance, promoted by the United Kingdom government, predicted that AMR could lead to 10 million death per year by 2050, causing an increase of the overall healthcare costs of USD 100 trillion [26]. The economic and sanitary burden of AMR is thus huge [27,28]. A recent systematic analysis by the Antimicrobial Resistance Collaborators [29] developed predictive statistical models to analyze 471 million records covering 16 different countries around the world. The authors estimated the occurrence of 1.27 million (95% uncertainty interval: 0.911–1.71) deaths directly imputable to AMR and 4.95 million (95% uncertainty interval: 3.62–6.57) deaths associated with bacterial AMR only in 2019. Particularly, intra-abdominal infections alone resulted as the third cause of AMR-related death [29]. The economic impact is also devastating, leading to an additional cost of USD 20 billion in the US every year [30].

The global burden of AMR calls for a holistic reaction. Since the discovery of penicillin more than 90 years ago, germs have increasingly developed new AMR patterns, while antimicrobial development and production has slowed down. The optimal management of AMR requires several steps to preserve the effectiveness of existing antimicrobials by preventing, detecting, containing, tracking, and treating, AMR infections across health care, agriculture, and animal welfare. In 2015, Tillotson et al. suggested four interventions to attenuate AMR impact on healthcare systems: (1) preservation of the effectiveness of existing and future antibiotics; (2) reduction of antimicrobial over-prescribing and misuse, favoring timely prescription of effective antibiotics at adequate dose when indicated; (3) development of new antibiotics; and (4) the implementation of economic policy on global AMR funds and source distribution [31]. AS and IPC represent two crucial cornerstones to fight against AMR and minimize its impact on patients, the healthcare system, the economy, and the environment. Since 2002, the US Centers for Disease Control and Prevention (CDC) has promoted a campaign against AMR based on four concomitant tactics: infection prevention; punctual and rapid diagnosis and treatment; the judicious use of antimicrobials; and the prevention of transmission [32]. Surveillance is a key component of the AMR control program through the identification of emergent multidrug-resistant microbes, the tracking of AMR epidemiological variations, and the analysis of interventional outcomes [32]. Additionally, educational interventions with the aim of improving awareness and knowledge of AMR was proved to be effective in decreasing the healthcare transmission of multidrug-resistant bacteria [33,34,35,36].

## 3. Inflammatory Response and Infections in Acute Care Surgery: The Evolution of Sepsis Definition and the Role of Biochemical Markers and Prediction Scores

Inflammation is an indispensable response of the immune system induced by exogenous or endogenous detrimental insults, namely infections, traumas, surgery, malignancies, and many others (Table 1) [37].

Despite its physiological nature, the dysregulation of the inflammatory response can trigger a chain of events that culminate in tissue and organ injury. Damage-associated molecular patterns and pathogen-associated molecular patterns represent some of the immune-stimulators promoting this pathologic inflammatory response [38,39].

In 1991, the American College of Chest Physicians and the Society of Critical Care Medicine coined the definition of systemic inflammatory response syndrome (SIRS) and a new definition of sepsis [40]. SIRS was defined by the presence of at least two out of the following four criteria: body temperature over 38 °C or under 36 °C; heart rate greater than 90 beats/min; respiratory rate greater than 20 breaths/min or partial pressure of CO_2_ less than 32 mmHg; and white blood cell count greater than 12,000/mm^3^ or less than 4000/mm^3^ or over 10% immature forms or bands. Sepsis was defined as a SIRS promoted by a confirmed infectious process [40].

More recently, the release of Sepsis-3 by the European Society of Intensive Care Medicine, and the Society of Critical Care Medicine in 2016, has re-defined sepsis as a severe organ dysfunction triggered by a dysregulated host response to infectious agents [41]. This newer definition presupposed the use of the sequential organ failure assessment (SOFA) criteria for sepsis diagnosis [41]. Conversely, the SIRS criteria were forwarded due to their high sensitivity and low specificity [41,42]. Indeed, variations in temperature, white blood cell count, respiratory and heart rate, do not always indicate a pathologic inflammatory response and represent common findings in hospitalized patients, including those without infectious diseases [43,44,45,46,47]. To note, the clinical presentation of infectious and noninfectious systemic inflammatory responses could overlap significantly and that almost all infectious diseases promote SIRS, but not all SIRS have an infectious origin. Surgery itself may result into a proinflammatory state, often revealing a physiological and pertinent adaptive host response [48,49,50,51,52]. Specifically, the type of surgical technique (open vs. minimally invasive) [53,54,55,56,57], operation time [58], blood transfusion [59,60], and anesthesia techniques [61,62], may alter the immune system and amplify the systemic inflammatory response. The modern surgeon should develop the ability to ascertain the magnitude of postoperative inflammation, distinguishing between beneficial from pathologic host response, and between infectious and non-infectious etiology. Despite the potential improvement of postoperative outcomes due to the steroid modulatory effect on systemic inflammatory response [63,64], SIRS remains a clinical manifestation of a triggering insult and its management implies the treatment of the primary causal etiology.

Nowadays, the use of supplementary diagnostic tools such as C-reactive protein (CRP), procalcitonin, and an early warning score, could facilitate the early diagnosis of adverse events and time-sensitive interventions in surgical patients with suspected infection [65]. CRP is a non-specific but high-sensitive indicator of inflammatory response, useful to predict postoperative complications [66,67,68]. CRP concentration and fluctuation over time were proved effective as a negative predictive test for anastomotic leakage [69,70], and postoperative infectious complications [71,72,73,74,75]. Besides, procalcitonin may be a valuable marker in the assessment of infections in trauma and ACS patients [76,77,78,79]. Evidence from a meta-analysis of 3244 reports showed the efficacy of procalcitonin for the early identification of sepsis in critically ill patients, resulting into a pooled sensitivity and specificity of 77% and 79%, respectively [80]. Furthermore, several randomized controlled trials evaluated the use of procalcitonin to guide antimicrobial duration in septic patients [81,82,83,84]. According to a multidisciplinary task force of experts, procalcitonin may be considered as an excellent parameter for guiding treatment and duration of antibiotic therapy in general and emergency surgery [85]. Further, the use of an early warning score, namely quick SOFA (qSOFA) [41,79,86,87,88], SIRS [89,90,91], Modified Early Warning Score (MEWS), and National Early Warning Score (NEWS) [92], can facilitate the stratification of risk in patients with suspected infection. Churpek et al. compared the accuracy of qSOFA, SIRS, MEWS, and NEWS, by analyzing a cohort of 30,677 non-ICU patients with suspicion of infection in emergency department and hospital wards. MEWS and NEWS resulted more accurate than qSOFA and SIRS scores to predict in-hospital mortality and ICU admission, with the NEWS being the most precise tool [46]. A recent meta-analysis including 42,623 patients from seven studies pointed out that Machine Learning (ML) had a higher predictive performance for hospital-acquired sepsis than SOFA, SIRS, and MEWS (the pooled area under the receiving operating curve was 0.89 for ML, 0.78 for SOFA, 0.7 for SIRS, and 0.5 for MEWS) [93]. Artificial intelligence emerged in the digital age as a promising technology for healthcare data analysis and interpretation, being able to support clinical decision-making, risk assessment, and pre-operative planning, also in the ACS setting [94,95,96,97,98,99]. Bertsimas et al. designed an interactive and user-friendly calculator, called POTTER (Predictive OpTimal Trees in Emergency Surgery Risk), based on non-linear machine learning techniques applied to a cohort of 382,960 patients; it appears to reliably predict 30-day postoperative mortality, morbidity, and 18 specific complications (with the higher performance in predicting septic shock) in emergency surgery patients [100]. All these predictive tools may represent valuable tools for acute care surgeons who often have to face complex decisions.

## 4. The Management of Infections in Acute Care Surgery

The global public health challenge driven by infections in ACS refers either to patient morbidity, mortality, and quality of life, and further to financial and economic reverberations. In this perspective, the impact of HAIs and IAIs is wide.

HAIs are infections developed 48 hours or more after hospital or other healthcare facility admission, or within 30 days after having received health concerns [101]. HAIs encompass surgical site infections (SSI), catheter-associated urinary tract infections, central line-associated bloodstream infections, Clostridium difficile infections, and ventilator-associated pneumonia, among many others. Acute care facilities and ACS populations represent boost sanctuaries for emergence, development, and transmission of infections and multi-resistant organisms. The use of medical devices, such as ventilators, central venous catheters, intra-abdominal drains, and urinary catheters, but also the surgery itself, should be considered as significant risk factors for HAI acquisition. Despite the considerable predictability of HAIs, their epidemiological weight is significant. In 2011, the World Health Organization (WHO) reported that HAIs affect 7% of hospitalized patients in developed and 10% in developing countries, leading to a burden of 4 million cases in Europe and 1.7 million in US with 39,000 and 99,000 directly attributable deaths, respectively [102]. SSIs are the most common HAI in surgical units with a major impact in terms of morbidity, mortality, length of hospital stay, and costs [103]. In a multicentre and international cohort study, SSIs were identified in 12.3% of patients within 30 days after gastrointestinal surgery and disease incidence fluctuated significantly between countries with high, middle, and low-income (9.4% vs. 14.0% vs. 23.2%, respectively, *p* < 0.001) [104]. Additionally, Magill et al. highlighted that AMR patterns were detected in up to 60% of the microbes isolated from infected surgical sites [105].

IAIs are common surgical emergencies and represent an important cause of morbidity and mortality worldwide. In the modern era, the IAIs-related mortality has dropped sharply compared to the values higher than 50% reported in the middle of the last century [106]. Indeed, Sartelli et al. reported an overall 9.2% mortality rate in a multicenter cohort of 4553 patients with complex intra-abdominal infections over 4 months [107]. Moreover, the presence of severe sepsis considerably increased IAI mortality rate [108,109,110]. A multicentre observational study on ICU patients with a confirmed IAI diagnosis attested that 68% of IAIs were hospital-acquired, with an overall AMR prevalence of 26.3%, and overall mortality of 29.1% directly related to infection severity [111]. IAIs include appendicitis, bowel and colorectal perforations, cholecystitis, diverticulitis, gastroduodenal perforations, bowel and colorectal ischemia, necrotizing pancreatitis, and many other diseases. Despite the wide spectrum of conditions, IAIs are usually classified into uncomplicated and complicated. Uncomplicated IAIs refer to infections limited to a hollow viscus, whereas complicated IAIs presuppose the involvement of a naturally sterile area of the abdomen, such as the peritoneal cavity, other abdominal organs, abdominal wall, mesentery, and retroperitoneum [112]. Differently, the US Food and Drug Administration defined the latter as disorders requiring SC procedures, with the aim to better clarify the definition and identification of complicated IAIs. In principle, uncomplicated IAIs can be handled with either surgical SC or with antibiotics alone, while complicated IAIs require both SC and antibiotic therapy [113]. For instance, complicated IAIs may include secondary or tertiary peritonitis, intra-abdominal abscesses, or intra-abdominal phlegmons.

Worldwide, hospital and patient cares require comprehensive and standardized policies, procedures, and strategies, to minimize the impact of HAIs and AMR, and to optimize IAI treatment. AS, IPC, and SC, are the three synergistic cornerstones of the multidisciplinary management of infections in acute care facilities (Figure 1).

### 4.1. Source Control

SC in ACS refers to any intervention necessary to identify and eliminate the sources of infection and finally to restore normal physiological balance [114]. Combined with targeted antibiotic therapy, it represents one of the fundamentals of IAI care [113]. Delayed and partial interventions may significantly worsen SC outcomes in IAI patients; thus, prompt and goal-directed tactics are pivotal concepts to achieve the maximal efficiency of SC [106]. Sartelli et al. recently defined the four rules of an ideal SC [113]:(First) Time: start as soon as possible;Totalization: remove any infective source;Technique: use adequate techniques;(Second) Time: avoid clinical deterioration through required or planned successive procedures.

The first cornerstone is thereby the control of foci of infection within the shortest delay possible, especially in critically ill patients [115]. Elsewise, the optimal timing of IAI SC has long been debated [116]. Azuhata et al. showed the negative impact of each hour delay on survival in patients with gastrointestinal perforation and septic shock [117]. The 2017 Surgical Infection Society Revised Guidelines indicated that SC interventions must be undertaken within 24 hours from the diagnosis of IAI, except in the case of clinical evidence supporting non-interventional or delayed approach, as appropriate (strength of the recommendations: Grade 2-B), and in the case of sepsis or septic shock always seeking immediate treatment (strength of the recommendations: Grade 2-C) [112]. The same suggestion emerged from a comprehensive European review [118] and an international multi-society document on IAI [113]. According to the Surviving Sepsis Campaign guidelines, any specific anatomic diagnosis of infection requiring emergent SC should be rapidly identified or excluded and any essential SC intervention should be implemented [79].

The second cornerstone is the completeness of SC. Several studies highlighted the association between incomplete interventions and severe adverse outcomes [116,119,120,121]. Indeed, the persistence of infected fluid or contaminated tissue may nullify the benefits of resuscitation and antimicrobial therapy and prevent the physiological recovery. Van de Groep et al., investigating on a large cohort of critically ill patients with IAI, reported approximately 50% of infection persistence or recurrence after the first SC with a median of three procedures per patient and 67% SC adequateness on day 14 [122]. SC failure may occur in more than one-in-five patients [112] and in this scenario a prompt abdominal re-exploration should strongly be considered [106,113].

The selection of the optimal SC intervention should consider the risk-benefit ratio of the procedure, the site and severity of the infection, the patient’s fitness and health state, the medical expertise, and the surgical, interventional, or diagnostic staff availability. SC encompasses many procedures, namely drainage of abscess or infected collection, debridement of infected or necrotic tissue, removal of an infected device, and definitive control of the microbial source (e.g., resection of infected organs–appendicitis, cholecystitis, bowel ischemia – and suture or resection of perforated viscus–gastric and duodenal perforation, perforated diverticulitis). The international WISS study pointed out the source of infection in 4553 patients from 132 hospitals over a 4 month period, as follows: 34.2% appendicitis, 18.5% cholecystitis, 11% gastroduodenal perforations, 8.5% postoperative, 5.9% non-diverticular colonic perforation, 5.4% small bowel perforation, 5.2% diverticulitis, 2.5% post-traumatic perforation, 1.1% pelvic inflammatory disease, and 7.7% other [107].

Techniques to achieve SC encompass primary and alternative interventions which should be selected according to severity and stage of the disease, patients’ characteristics, and hospital resource availability (Table 2).

The primary goal of SC interventions is to identify the origin of IAI and to control the cause of abdominal sepsis. Peritoneal fluid cultures should be reached during SC procedures with the purpose of guiding antimicrobial therapy on the basis of antibiogram. Recent evidence reported the growing identification of multidrug-resistant microbes in IAI, thereby AMR may represent a determining factor for SC failure [134,135,136]. In the setting of complicated IAIs, a short-course antimicrobial therapy following an adequate SC intervention is a proper strategy for complicated IAIs, while it is not required for uncomplicated IAIs, such as uncomplicated cholecystitis or appendicitis [106,137,138,139,140]. The empiric antimicrobial therapy for complicated IAIs should be started as soon as possible and drug choice should be based on local ecology and AMR data, preferring antimicrobials with a spectrum of action against Enterobacteriaceae, enteric streptococci and obligate enteric anaerobes [79,106,112,113,119,141]:Amoxicillin/clavulanate should be considered for empiric therapy in accordance with local AMR epidemiology, because the emergence of extended-spectrum beta-lactamases (ESBL) producing Enterobacteriaceae has reduced its efficacy [142,143,144,145,146];Piperacillin/tazobactam is considered the optimal option for the treatment of complicated IAIs due to its broad spectrum of efficacy against Enterobacteriaceae, Pseudomonas, anaerobes, non-resistant Enterococci and certain classes of ESBL [145,147,148];Third-generation cephalosporines (e.g., cefotaxime, ceftriaxone, ceftazidime) in combination with metronidazole are active against Enterobacteriaceae and may be considered for uncomplicated IAIs [146,149]. Recently, two fifth-generation cephalosporines, namely ceftolozane/tazobactam and ceftazidime/avibactam, have been approved as treatments for complicated IAIs in combination with metronidazole, given to their action against several multidrug resistant bacteria [150,151,152];Carbapenems (imipenem, meropenem, ertapenem) have a broad spectrum of action and represent a useful resource against ESBL [146,153,154];Aminoglycosides (e.g., amikacin, tigecyclin) should be considered in case of beta-lactam allergy due to controversies on their toxic side effects [106,112,146,155].

Empirical antimycotic therapy may be appropriate in selected conditions (hospital-acquired IAIs, immunosuppressive state, critical illness) [106,156].

According to the Surgical Infection Society, microbiologic data and antibiogram can be used to reassess and optimize antimicrobial therapy [112]. Furthermore, the adequateness and appropriateness of antimicrobial treatment should be re-thought daily [157]. Concerning surgical SC, the usefulness of a prophylactic drain placement after digestive surgery has long been debated [158,159,160]. Intra-abdominal drains paradoxically may promote surgical site infections [161] and increase hospital length of stay and costs [162,163] in several emergency scenarios, namely acute appendectomies, or cholecystectomies. De Waele et al. recommended a limited duration of abdominal drains in the treatment of abdominal sepsis, a prompt drain removal once the source has been controlled, and to avoid culturing drains upon removal [164].

In principle, less invasive and more effective procedure should be chosen to reach SC [112]. During the last decades, minimally invasive surgery has gained wider acceptance in the treatment of IAI and abdominal sepsis ensuring concomitant advantages of diagnostic and operative tools with better post-operative outcomes (e.g., decreased postoperative infections and pain, shorter hospital stay and earlier recovery of physiological functions). Indeed, several authors recognized a laparoscopic and robotic approach as feasible, effective, and safe techniques for many emergency conditions [165,166,167,168]. Despite the clear advantages, minimally-invasive surgery in acute care setting requires adequate technical skills and specialized training.

### 4.2. Infection Prevention and Control

Infection prevention and control (IPC) is a pragmatic, evidence-based approach, which primary goal is preventing patients and health workers acquisition of HAIs, reducing the spread of HAIs within health care facilities, and weakening AMR [169]. For this purpose, the WHO issued several recommendations, defined as core components [169], and their respective minimum requirements [170] to ensure the effectiveness of IPC programs at national and facility level. The implementation of IPC core components should be faced using a stepwise approach [171], according to the following five steps:Preparing for action;Baseline assessment;Developing and executing an action plan;Assessing impact;Sustaining the program over the long term.

IPC is a multidisciplinary strategy that involves all levels of the health system and different hospital professionals. According to the systematic review by Zingg et al., an adequate IPC programme in an acute care hospital must include as a minimum standard at least one dedicated and specialized infection-control nurse, a dedicated physician trained in IPC, and microbiological and data management support [172]. A recent worldwide cross-sectional survey on this topic revealed that a multidisciplinary IPC team composed by a median of six professionals (including most frequently microbiologists 72.4%, infectious diseases specialists 70.2%, nurses 68.4%, pharmacologists 67.6%, and surgeons 56.7%) was present in about 90% of participating hospital [173].

The effective implementation of infection control programs requires the engagement of well-designed organizational and structural models: specifically, ward occupancy have not to exceed its planned capacity; health-care workload must be adapted accordingly; and dedicated full-time nurses and physicians should be preferred over pool or agency professionals [172]. Indeed, several studies showed the association between HAIs and bed occupancy, high workload, and pool or agency professionals [174,175,176,177,178]. Furthermore, the usability of IPC programs is closely related to the development of multimodal strategies for team education and training, allowing the transposition of IPC pivotal fundamentals into documents and protocols based on local context and facilitating the good daily practice through surveillance, auditing, and personal feedback [172].

SSIs account for most HAIs among surgical patients. Additionally, SSIs are associated with longer hospitalization, higher costs, morbidity, and mortality [103], thus SSI prevention has a crucial role in all surgical departments worldwide. During the last years, the WHO [179,180,181], the US CDC [182] and the National Institute for Health and Care Excellence (NICE) [183], published evidence-based guidelines for the prevention of SSIs, dispensing recommendations for both preoperative, intraoperative, and postoperative surgical phase. The common goal of all these health institutions is to standardize patient care through the identification of the best practice, so promoting a favorable fallout on morbidity, mortality, and costs. The most recent WHO strong recommendations are reported in Table 3.

The potential predictability of SSIs and the detrimental impact of infections on patient care was the driving factor for numerous researches. For instance, the ongoing ROSSINI 2 trial (NCT number: NCT03838575) is a phase III, multicentre, multi-arm, and multi-stage study, designed to evaluate the clinical effectiveness of three in-theatre interventions (2% alcoholic chlorhexidine skin prep vs. iodophor-impregnated incise drapes vs. gentamicin-impregnated sponge) to decrease 30-day postoperative SSI after abdominal surgical incisions. At last, the recent position paper of the World Society of Emergency Surgery granted versatile recommendation on intraoperative prevention of SSI in patients with IAI in the emergency setting (Table 4).

During the last years, the US CDC has provided the following standard precautions for IPC in all patients in the healthcare setting [185]:Perform hand hygiene;Use personal protective equipment in case of possible exposure to infectious agents;Follow respiratory hygiene;Guarantee adequate patient placement;Clean and disinfect environment, patient equipment and medical devices adequately;Handle textile and laundry with caution;Follow safe injection practices;Allow healthcare professional safety;Employ transmission-based precautions in case of known or suspected infections.

Among US CDC guidelines for isolation precautions, education and training play a pivotal role [186]. The implementation and diffusion of educational strategies should be warranted through job- or task-specific programs, periodical update courses, incentives to involve all workers and beyond, and periodic assessments on staff competences [186].

### 4.3. Antimicrobial Stewardship

Antimicrobial stewardship (AS) is an emerging strategy conceived to reduce the AMR spread and to promote the judicious use of antimicrobials. It has been estimated that about a third of all antibiotics used in US acute care hospitals are unjustified and inappropriate [187,188] and antibiotic-related adverse events occur in one out of five hospitalized patients [189]. Thus, optimizing the adequate use of antimicrobials is a pivotal element for global health and patient safety. According to AS fundamentals, therapy effectiveness and adequateness should be achieved using the minimum number of antimicrobials for the shortest length of therapy at the appropriate dose ensuring the best ratio efficacy-complications.

AS programs represent a valuable tool for clinicians to standardize clinical practice and empower antibiotic use. Dortch et al. demonstrated a significant reduction of multidrug resistant gram-negative bacteria incidence and the concomitant decrease in broad spectrum antibiotic use after the implementation of hospital AS protocols in surgical and trauma intensive care units [190]. A retrospective study by Sartelli et al. analyzed the positive impact of AS programs implementation into a surgical unit on antimicrobial prescriptions and consumption. Comparing the pre-intervention and post-intervention periods, the mean monthly antimicrobial consumption decreased by 18.8%, passing from 1074.9 defined daily doses per 1000 patient days to 873 defined daily doses per 1000 patient days [191]. Additionally, three systematic reviews and meta-analysis supported a significant association between AS practices and reduction of incidence of infections and colonization with antibiotic-resistant bacteria and C difficile, suboptimal therapy, antibiotic-related adverse effects, length of hospital stay and hospital costs [192,193,194]. From a world cross-sectional survey emerged that local AS protocols were applicated in three out of four hospitals (76.6%), by introducing strategies to reduce the duration of therapy in 85.8% of cases or suggest alternative dosing tactics in 79.8% [173].

In 2007, the Infectious Diseases Society of America and the Society for Healthcare Epidemiology of America published guidelines for the development of hospital stewardship programs, identifying two principal recommendations for an effective AS: first, formulary restriction and preauthorization process for antimicrobial use was considered an useful option; and, secondly, the authors highlighted the need of prospective audit on antimicrobial use with intervention and feedback to the prescriber [187]. The greater short-term and immediate utility of restrictive interventions over persuasive recommendations was also showed by other studies [193]. More recently, the US CDC released the updated version of the “Core Elements” to guide AS strategies in hospital settings [195], through the following strategies:Dedicate human, financial, and information technology resources to achieve program effectiveness;Designate a leader or co-leaders (e.g., physicians or pharmacists) accountable for program development and results;Designate a pharmacist able to improve antimicrobial use;Implement in-hospital interventions (e.g., prospective audit and feedback or preauthorization);Monitor antibiotic use, impact of interventions and infection-related outcomes;Report information on antibiotic use and resistance to dedicated hospital workers;Educate hospital workers and patients on antibiotic-related adverse reactions, AMR, and optimal antimicrobial use.

The WHO provided 10 similar interventions to promote a highly effective AS, differentiating them into initial (prior or at the time of a prescription) or subsequent (after a prescription) steps in clinical decision-making [196]. Furthermore, this practical WHO guide offered a flexible, step-up approach base on institution’s resources (e.g., timely access to microbiologic data and availability of clinical pharmacists or dedicated AS team), so AS interventions indicated for limited-resource facilities are lesser [196]. Consequently, AS requires a dynamic balance between antibiotic selection, adequate dose, and duration of therapy, and ultimately an optimal AS program should be based on international/national antimicrobial guidelines and tailored to AMR ecosystem, local microbiology, and hospital resources. In this scenario, the AS multidisciplinary team may constitute the trait d’union between global recommendations and local protocols, guiding clinicians in antibiotic prescribing practices.

The crucial role of AS in ACS setting is even more important, as antimicrobials are used for both surgical prophylaxis and the management of HAIs and IAIs. Leeds et al. evaluated the antibiotic-prescribing practices for HAI into a colorectal surgery unit, reporting a treatment deviation (e.g., antibiotic selection, duration) from international guidelines and evidence-based recommendations in 73% of cases [197]. In details, the treatment deviation was related to antibiotic selection and therapy duration in about 25% and 60% of patients, respectively [197]. Therefore, the authors advocated the consolidation of AS principles in surgical practice in order to optimize antimicrobial use, selection, and duration.

## 5. Conclusions

Infections in ACS represent a significant cause of morbidity and mortality worldwide. The optimal management of infections in ACS setting requires a dynamic balance between a goal-direct SC strategy, effective and evidence-based IPC-AS programs, and an adequate antimicrobial use. We conceived this review to provide a snapshot of the available international and national SC, IPC, and AS recommendations as a useful tool for the acute care surgeon. Indeed, the modern acute care surgeon plays a key role in the management of infections and in the prevention of AMR, so the ACS skills should be optimally oriented to the prompt evaluation and management of several time-sensitive surgical and infectious diseases. In the ACS scenario, the surgeon’s decision-making may comprehend the extensive knowledge and the successful application of SC, IPC, and AS principles. A recurring theme of 21st century is the centralization of health care into high-specialized centers. This paradigm should be overturned in the framework of infections in ACS, allowing a standardization of care in every surgical facility worldwide, passing from high-volume to low-volume centers or from high-income to low-income countries. The selection and timing of antimicrobial therapy and SC tactics should be tailored to the patient’s characteristics, always considering international and national guidelines, local ecology, and resistance data. 

## Figures and Tables

**Figure 1 antibiotics-11-01315-f001:**
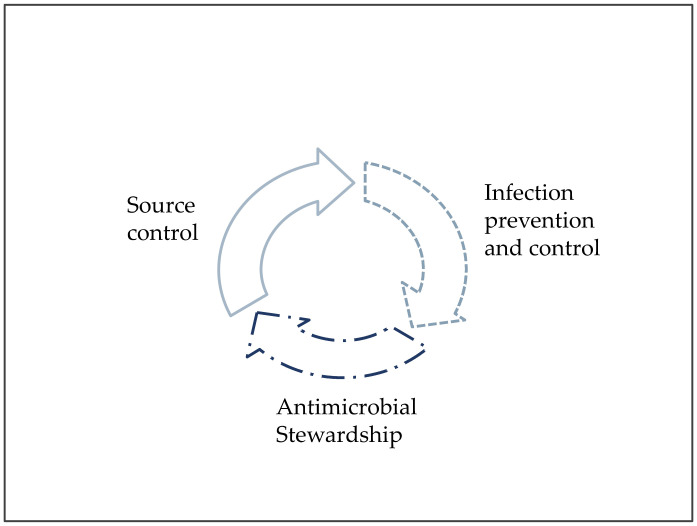
The three pillars of patient care against infections and antimicrobial resistance.

**Table 1 antibiotics-11-01315-t001:** Infectious and non-infectious causes of inflammatory host response.

Infective	Non-Infective
Pneumonia	Trauma
Endocarditis	Surgery
Skin and soft tissue infections	Burns
Urinary tract infections	Malignancies
Intra-abdominal infections	Autoimmune disorders
	Cirrhosis
	Hemorrhage
	Adrenal insufficiency
	Acute pancreatitis
	Acute ischemia
	Acute aspiration

This table contains a partial list of the most common etiologies of inflammatory host response.

**Table 2 antibiotics-11-01315-t002:** Source control strategies for most common intra-abdominal infection scenarios.

Primary Intervention	Alternative Intervention
***Acute appendicitis*** [115,123,124,125]	
Appendectomy via laparoscopy is the preferred approach for both uncomplicated and complicated (perforation and peri-appendiceal abscess) acute appendicitis	The antibiotic-first strategy is a feasible option in selected patients with uncomplicated acute appendicitis illustrating the risk of failure and misdiagnosis Percutaneous drainage could be a reasonable option in case of appendicitis with peri-appendiceal abscess and phlegmon
***Acute cholecystitis*** [126,127,128]	
Laparoscopic cholecystectomy is recommended for source control in case of timeliness clinical onset (<7–10 days), peritonitis or sepsis Subtotal cholecystectomy is an alternative technique for advanced inflammation, gangrenous gallbladder, or difficult anatomy	Antibiotics and delayed cholecystectomy (beyond 6 weeks from the first clinical onset) are preferred in non-adequate clinical onset (>7–10 days) Percutaneous cholecystostomy drainage or endoscopic transpapillary gallbladder drainage or ultrasound-guided transmural gallbladder drainage should be considered as surgical alternatives in patients unfit for surgery (e.g., severe comorbidities or septic shock)
***Gastroduodenal perforation*** [129]	
Operative source control is indicated in presence of significant pneumoperitoneum or extraluminal contrast extravasation or signs of peritonitis Laparoscopy in the preferred approach in stable patients Primary repair is recommended for perforated peptic ulcer smaller than 2 cm Resection or repair ± pyloric exclusion ± external bile drainage should be considered in case of perforated peptic ulcer > 2 cm Resection with contextual operative frozen pathologic examination is advisable in case of malignancy suspicion	Non-operative management could be considered in extremely selected cases with sealed perforation confirmed on water-soluble contrast imaging. It consists of the association of absolute fasting + antibiotics + decompression via nasogastric tube + proton pump inhibitor therapy and requires a follow-up endoscopy at 4–6 weeks.
***Postoperative peritonitis*** [113]	
Prompt surgical management is mandatory in case of diffuse peritonitis	Antibiotics + percutaneous drainage should be considered for localized intra-abdominal abscesses in stable patients with no signs of generalized peritonitis
***Small bowel perforation*** [113]	
Primary repair is an option in selected patients with minimal peritoneal contamination and small defect, while bowel resection ± anastomosis is indicated in remaining cases Stoma creation or perforation exteriorization is indicated in case of critical illness or severe inflammation/peritonitis and edema of the bowel	In case of small bowel ischemia, resection and delayed anastomoses should be considered as alternative
***Diverticulitis*** [130,131,132]	
Oral or intravenous antibiotics are indicated in hemodynamically stable patients without drainable collection, pericolic extraluminal gas or small (< 4–5 cm) diverticular abscess. It should also be considered in case of CT findings of distant free gas without diffuse intra-abdominal fluid, only if a close follow-up can be guaranteed Primary resection with anastomosis with or without diverting stoma is recommended in clinically stable patients without comorbidities	Hartmann’s procedure is recommended for 1) diffuse peritonitis and critically ill patients in case of hemodynamically stability; 2) frail patients with sepsis and temporary hemodynamic instability returning to normal pressure after crystalloid infusion Damage control surgery is indicated for hemodynamic instable patients unresponsive to fluid administration and an open surgical approach is mandatory Percutaneous drainage is recommended in hemodynamically stable patients with large abscesses
***Esophageal perforation*** [133]	
Antibiotics + absolute fasting + proton pump inhibitor therapy can be indicated in stable patients with early onset, minimal esophageal damage, and contained contamination, if highly specialized surveillance is guaranteed	Surgery should be advised in absence of non-operative management criteria (e.g., hemodynamic instability)
***Acute cholangitis*** [113]	
Antibiotics + biliary drainage via endoscopic retrograde cholangiopancreatography (treatment of choice) or percutaneous biliary drainage (in case of ERCP * failure) are the first-line approach	Open drainage should be considered only in case of failure or contraindication to endoscopic or percutaneous interventions

* ERCP: endoscopic retrograde cholangiopancreatography.

**Table 3 antibiotics-11-01315-t003:** WHO strong recommendations for the prevention of surgical site infection *.

Core Topics
Perioperative *intranasal applications of mupirocin 2%* ointment with or without a combination of chlorhexidine gluconate body wash is indicated in patients undergoing cardiothoracic and orthopedic surgery with known nasal carriage of Staphylococcus aureus
*Surgical antibiotic prophylaxis* should be administered within 120 minutes before surgical incision, even though considering antibiotic half-life
*Mechanical bowel preparation* should not be used for the prevention of SSI in elective colorectal surgery if not associated with preoperative oral antibiotics
*Hair removal* and shaving is strongly discouraged in any surgical procedure. If inevitable, hair should be removed only with a clipper
Alcohol-based antiseptic solutions based on chlorhexidine gluconate are recommended for *surgical site skin preparation* before surgery
*Surgical hand preparation* should be performed by scrubbing with either a suitable antimicrobial soap and water or using appropriate alcohol-based solutions before wearing sterile gloves
Prolongation of *surgical antibiotic prophylaxis after the surgical procedure* is not indicated for the purpose of preventing SSI

* Only strong recommendations are reported in the table.

**Table 4 antibiotics-11-01315-t004:** WSES recommendations for the prevention of surgical site infection presented with the specific statements and their GRADE of supporting literature (indicated with superscript numbers).

Statements	GoR
Continuous and interrupted skin suture present no significant difference in terms of SSI incidence, but superficial wound dehiscence is lower in continuous suture	1B
Triclosan-coated suture significantly reduce SSI prevalence	1B
The efficacy of intraperitoneal or topic wound irrigation with antibiotics in preventing SSI is not supported by sufficient data	2B
The efficacy of incisional wound irrigation with saline or povidone solution before skin closure in preventing SSI is not supported by sufficient data	2B
The use of wound protectors reduces incisional SSI ^1^, in particular the effectiveness of dual-ring devices is superior to single-ring ones ^2^	^1^ 1A ^2^ 1B
The role of plastic adhesive drapes with or without antimicrobial properties in preventing SSI is not supported by sufficient data	2C
The role of subcutaneous drainage of incisional wounds before closure in preventing SSI is not supported by sufficient data	2B
The role of double gloving in preventing SSI is not supported by sufficient data ^1^. Change of gloves at time intervals during surgery may be beneficial ^2^	^1^ 2B ^2^ 2C
Negative-pressure wound therapy may be a valid option for SSI prevention, especially in SSI high-risk patients (e.g., contaminated, and dirty, surgical wounds)	2C
Intraoperative normothermia prevents SSI ^1^ and active warming devices are useful in achieving normothermia and reducing SSI ^2^	^1^ 1A ^2^ 1B
Perioperative hyperoxygenation does not reduce SSI	2B
Delayed primary skin closure may reduce SSI incidence ^1^ and it should be considered in high-risk patients underwent contaminated abdominal surgery ^2^	^1^ 2B ^2^ 2C
Additional intraoperative antibiotics may be administered in case of emergency surgery for IAI always considering drug pharmacokinetic and pharmacodynamic	1C

WSES: World Society of Emergency Surgery; GoR: Grade of Recommendation; SSI: Surgical site infection; IAI: intra-abdominal infection. Statements were evaluated according to the Grading of Recommendations, Assessment, Development and Evaluation (GRADE) [184].

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
