# Peer review of "Antimicrobial Challenge in Acute Care Surgery"

_antibiotics, 2022, doi:10.3390/antibiotics11101315_

Round 1

Reviewer 1 Report

The authors wrote a comprehensive review article on antimicrobial challenge in acute care surgery highlighting several aspects of infection prevention, control, to antibiotic stewardship. I have a few comments:

1. The review article lack originality but make up for its comprehensive coverage of the topic. 

2. The section on SIRS is a bit elementary, and somewhat redundant in the explanation, especially to the journal's highly professional readers.

3. The majority of patients present with acute surgical conditions are in a SIRS/septic state. I find the section 3 of the article somewhat bothersome with the findings that patients presented with SIRS has higher risk of morbidities, and adverse outcomes. While that is true based on supporting literature, SIRS is a manifestation of patients' illness and not the cause. So, to associate SIRS to adverse outcomes without defining the causes is a bit misleading. 

4. Table 3 is confusing with operative vs non-operative source control for intra-abdominal infection since the treatments are usually not equal in benefit to patients. Instead, the authors should consider re-organize the table to preferred vs alternative treatment category. 

5. The authors should also consider including the CDC's recommendations to infection prevention and control section 4.2 readers in the US.

Author Response

Manuscript ID: antibiotics-1920524

Antimicrobial challenge in acute care surgery

Dear Editor,

Thank you for considering our study. We carefully revised the entire manuscript according to the reviewers’ comments, which were very helpful to improve the article. We provide a point-by-point response letter below.

Response to the reviewers’ comments

The reviewer’s comment is indicated with R, Authors’ response with A

Reviewer: 1

Comments to the Author
R: The authors wrote a comprehensive review article on antimicrobial challenge in acute care surgery highlighting several aspects of infection prevention, control, to antibiotic stewardship. I have a few comments:

R1: The review article lack originality but make up for its comprehensive coverage of the topic.

A: We thank the reviewer for the time spent to analyze the review and for the suggestions that were highly helpful to improve our manuscript.

R2: The section on SIRS is a bit elementary, and somewhat redundant in the explanation, especially to the journal's highly professional readers.

R3: The majority of patients present with acute surgical conditions are in a SIRS/septic state. I find the section 3 of the article somewhat bothersome with the findings that patients presented with SIRS has higher risk of morbidities, and adverse outcomes. While that is true based on supporting literature, SIRS is a manifestation of patients' illness and not the cause. So, to associate SIRS to adverse outcomes without defining the causes is a bit misleading.

A: We thank the Reviewer for bringing our attention to this sensitive point. We have substantially revised the paragraph contents for clarity, so we hope that you will find it improved. We also acknowledge that the correlation of SIRS with a higher risk of morbidities was misleading, so we clarified this concept in the paragraph underlying that SIRS is a manifestation of patients' illness and not the cause.

R4: Table 3 is confusing with operative vs non-operative source control for intra-abdominal infection since the treatments are usually not equal in benefit to patients. Instead, the authors should consider re-organize the table to preferred vs alternative treatment category.

A: We thank the reviewer to have noticed this in Table 3, which was changed for clarity as suggested.

R5: The authors should also consider including the CDC's recommendations to infection prevention and control section 4.2 readers in the US.

A: We thank the reviewer for this useful suggestion. We added CDC's recommendations both in antimicrobial resistance and infection prevention and control paragraphs.

Reviewer: 2

R1: The summary of the literatures is not comprehensive, and part of the literatures needs to be updated.

A: We would like to thank the Reviewer for his constructive suggestions and comments. The summary of the literature has been updated as requested.

R2: The significance of this review is not highlighted in the abstract.

A: We thank the reviewer for this suggestion. We modified the abstract in order to better bring out all the aspects of this review.

R3: The keywords need be revised to cover the whole content of the manuscript.

A: We thank the reviewer for this suggestion. We added new keywords to cover the whole content of the manuscript.

R4: The review of measures taken in this manuscript is relatively few, which needs to be further supplemented.

A: We thank the reviewer for these valuable suggestions that were highly useful to improve the quality of our manuscript. The paragraph on systemic inflammatory response has been revised and new contents have been added in this study. We hope you will find it improved.

Reviewer 2 Report

1.  The summary of the literatures is not comprehensive, and part of the literatures needs to be updated.

2.  The significance of this review is not highlighted in the abstract.

3.  The keywords need be revised to cover the whole content of the manuscript.

4.  The review of measures taken in this manuscript is relatively few, which needs to be further supplemented.

Author Response

(The authors gave the same response as above.)

Round 2

Reviewer 1 Report

The authors have made appropriate revision based on prior comments. However, I recommend the authors to proofread the article for better word selection and sentence syntax for more elegant language description.

Author Response

Manuscript ID: antibiotics-1920524

Antimicrobial challenge in acute care surgery

Response to the Editor and Reviewers’ comments

The reviewer’s comment is indicated with R, Authors’ response with A

Editor

E: I appreciate the revision performed, but in the review there is a lack in the treatment of acute diverticulitis. I suggest to improve this topic with the following state of art and position papers: Cirocchi R, Sapienza P, Anania G, Binda GA, Avenia S, di Saverio S, Tebala GD, Zago M, Donini A, Mingoli A, Nascimbeni R. State-of-the-art surgery for sigmoid diverticulitis. Langenbecks Arch Surg. 2022 Feb;407(1):1-14. Nascimbeni R, Amato A, Cirocchi R, Serventi A, Laghi A, Bellini M, Tellan G, Zago M, Scarpignato C, Binda GA. Management of perforated diverticulitis with generalized peritonitis. A multidisciplinary review and position paper. Tech Coloproctol. 2021 Feb;25(2):153-165.

A: We thank the Editor for the time spent to analyze the revisions performed and for this suggestion regarding the treatment of acute diverticulitis. We added the recommended papers in the bibliography and tables, improving the text of the corresponding section.

Reviewer: 1

Comments to the Author
R: The authors have made appropriate revision based on prior comments. However, I recommend the authors to proofread the article for better word selection and sentence syntax for more elegant language description

A: We thank the Reviewer for this comment. We provided an English language revision before the article submission.

Reviewer: 2

R: The author has made careful revisions in accordance with the comments of the reviewers, and it can be accepted for publication.

A: We would like to thank the Reviewer for this comment.

Reviewer 2 Report

The author has made careful revisions in accordance with the comments of the reviewers, and it can be accepted for publication.

Author Response

(The authors gave the same response as above.)
